# Female Lower Body Muscle Forces: A Musculoskeletal Modeling Comparison of Back Squats, Split Squats and Good Mornings

**DOI:** 10.3390/jfmk9020068

**Published:** 2024-04-08

**Authors:** Jessica S. Jaeggi, Basil Achermann, Silvio R. Lorenzetti

**Affiliations:** 1Section Performance Sport, Swiss Federal Institute of Sport Magglingen (SFISM), 2532 Magglingen, Switzerlandacbasil@ethz.ch (B.A.); 2Institute for Biomechanics, ETH Zurich, 8092 Zurich, Switzerland

**Keywords:** biomechanics, musculoskeletal modeling, muscle forces, resistance training, ACL, EMG

## Abstract

The aim of this study was to analyze lower leg muscle forces during strength exercises such as back squats, good mornings and split squats, with a particular emphasis on females. By focusing on females, who are more vulnerable to anterior cruciate ligament injuries, we aimed to better understand muscle engagement and its role in injury prevention. Eight participants were monitored during exercises with a barbell load of 25% of body weight and, during the back squat, an additional 50% load. The analysis was conducted using personalized musculoskeletal models, electromyography (EMG) and Vicon motion capture systems to assess various muscle groups, including the m. gluteus maximus and m. gluteus medius, as well as the hamstring and quadriceps muscles. The back squat produced the highest forces for the quadriceps muscles, particularly the rectus femoris (>25 N/kg), as well as in the back leg during the split squat (>15 N/kg). The gluteal muscles were most active during good mornings and in the front leg of the split squat, especially the m. gluteus maximus medial part (>20 N/kg). The hamstrings generated the highest muscle forces in the front leg of the split squat, with the greatest forces observed in the m. semimembranosus. Our research highlights how musculoskeletal modeling helps us to understand the relationship among muscles, joint angles and anterior cruciate ligament injury risks, especially in strength training females. The results emphasize the need for personalized exercise guidance and customized models to make strength training safer and more effective.

## 1. Introduction

Resistance training has gained significant recognition as an effective method for enhancing physical fitness, optimizing athletic performance and promoting overall health and well-being. The proper selection of resistance levels and exercises can contribute to injury prevention by enhancing dynamic stability and preserving functional capacity [1,2,3]. Notably, females engaged in highly dynamic and jumping sports are more prone to anterior cruciate ligament (ACL) injuries compared to their male counterparts participating in similar sports. This increased susceptibility can be attributed to physiological and anatomical differences, which may have implications for knee stability and overall performance [4,5,6]. ACL injuries manifest when the mechanical load imposed on the ligament exceeds its capacity to withstand such forces. The knee ligaments primarily serve as stabilizers, while the surrounding muscles of the knee and hip, along with the gastrocnemius, play a secondary role in promoting optimal knee function [7].

Therefore, a number of studies have focused on neuromuscular training; internal loading conditions, such as muscle activation via electromyography (EMG); and patellofemoral joint forces [8,9,10,11,12,13,14], as well as external joint moments [15,16]. The aim of these studies was to uncover the inner workings of our bodies during resistance training. To effectively reduce the risk of injury, it is essential to fully understand the loading conditions affecting secondary stabilizers and to gain an understanding of both internal and external kinematics and kinetics during resistance training. Unfortunately, current non-invasive methods do not permit the direct measurement of forces during exercise performance [17]. However, musculoskeletal modeling techniques offers opportunities to unravel the complex interactions between the musculoskeletal system and external forces [18]. Such modeling approaches permit the assessment of joint forces, muscle activation and load distributions, thereby aiding in the identification of potential injury risks and the development of personalized training strategies.

Good mornings, split squats and back squats are multi-joint resistance exercises designed to target the lower limb muscles, including the gluteals, quadriceps and hamstrings, which contribute to knee stability as secondary stabilizers. These exercises enhance muscle strength, speed, power and dynamic stability and are considered comparable in terms of safety during execution [19]. However, the internal loading conditions and muscle forces associated with these exercises remain largely unexplored, especially in women. This gap in research extends to their biomechanical risk factors, including lower extremity muscle strength and activity, in relation to ACL injuries [20].

Consequently, the primary objective of this research was to investigate the variation in muscular forces across these exercises and to assess their alignment with measured EMG data. Additionally, we aimed to analyze the estimated forces at different knee angles to explore patterns of female muscular force and their implications for ACL injury risk factors.

## 2. Materials and Methods

### 2.1. Participants

Eight healthy female participants (mean age, 26.5 ± 3.1 years; mean height, 169.5 ± 5.5 cm; mean leg length, 90.6 ± 3.7 cm; mean weight, 69.1 ± 6.2 kg) with a minimum of 1 year of experience in strength training were recruited for this study. None of the participants had a history of neurological or motor deficits. They were instructed to wear their normal sports shoes with black shorts and a top. Prior to conducting the study, ethical approval was obtained from the relevant institutional review board. The study protocol complied with the Declaration of Helsinki for Human Experimentation and was approved by the regional ethics committee (Kantonale Ethikkommission Bern, Nr: 2021-00403). Informed consent was obtained from all participants, ensuring their voluntary participation and protection of their rights and privacy.

### 2.2. Data Collection

A VICON system comprising 10 cameras (200 Hz, Oxford Metrics Group, Oxford, UK) was used for 3D motion analysis. The plug-in gait marker set (from the VICON system), comprising 42 markers with a diameter of 16 mm, was applied by trained personnel along with two manual markers to track the barbell position. The Vicon cameras were arranged in a circular formation around the participants to minimize occlusion of the camera’s field of view by the weight plates. This configuration ensured optimal visibility of the participants and the markers for accurate motion analysis. Ground reaction forces were measured using two force plates (1000 Hz, Kistler AG, Winterthur, CH, Switzerland). In addition, eight surface EMG sensors were placed on specific leg muscles (gluteus medius, rectus femoris, vastus medialis, vastus lateralis, semitendinosus, biceps femoris, gastrocnemius medialis and gastrocnemius lateralis) to assess muscle activation patterns (Figure 1). The selection of muscles for EMG sensor placement was strategically focused on those essential for knee stabilization. This selection is supported by the literature emphasizing the importance of these muscles in maintaining knee joint stability and their involvement in common movements assessed in our study [7]. The standard Inverse Kinematik (IK) tool of OpenSim (v4.3) was used to calculate the joint angles, especially for the hip and knee. Using the minimal sum of weighted squared errors of the markers, the best position of the body for each frame is calculated to calculate the motion. The collected data served as reference data for scaling the models and running OpenSim simulations.

After a 3 min warm-up on a stationary bike, the participants engaged in standardized static and functional motion tasks to evaluate joint parameters. Barbell loads were adjusted based on body weight, with additional loads of 25% for split squat and good mornings and 25% or 50% for back squat. Each exercise comprised three cycles of five repetitions, enabling the subsequent calculation of average values for further analysis and evaluation. Participants were instructed to perform squats with a knee angle exceeding 90 degrees, ensuring the squat depth surpassed the parallel position. The range of motion for the other exercises was tailored to the individual capabilities of the participants. For additional technical details specific to the exercises discussed, please refer to the work by Schellenberg (2017) [21].

### 2.3. Modeling Simulation

The generic OpenSim model, specifically the Gait2392_Simbody version, was adapted for subsequent analysis [22,23]. In addition to the standard model, a barbell body with additional markers was included. Furthermore, six virtual markers were placed at the functional center of rotation (fCoR) of the hip, knee and ankle joints. These virtual markers, derived from the recalculated joint center locations of the plug-in gait model, allowed for subject-specific consideration of joint center positions and movements.

#### 2.3.1. Scaling

The scaling approach used in this study followed the procedures recommended by Schellenberg et al. [24]. The static trial position was used as the reference position of the joint angles. finSegment dimensions were determined based on relative distances between pairs of fCoRs, and the model markers were aligned with the captured markers in the static trial position. The markers were weighted differently, with a factor of 6 for bone and 1 for soft tissue markers and a factor of 100 for the pre-calculated center of rotation markers. This ensured accurate matching and alignment between the model and the captured data.

#### 2.3.2. Kinematics and Kinetics

Segment kinematics, including knee angles and hip angles, were calculated for each subject using the established procedures of OpenSim. The weighting of the markers was based on Schellenberg et al. [24], where the fCoR had a weight of 100. The start and end frames of the analyzed data were determined based on a barbell velocity threshold of greater than 40 mm/s [21]. Additionally, the individual kinetics were captured using OpenSim inverse dynamics, which were filtered with a frequency of 6 Hz. External forces from the force plates were incorporated into the analysis.

#### 2.3.3. Static Optimization

Muscle force and activation patterns of specific muscles were calculated by static optimization (Figure 1). At every static timeframe, the force–velocity relationship of muscle activation squared was set to equilibrium to generate an estimated activation pattern that best replicated the observed joint torques and kinematics [25].

### 2.4. EMG

EMG data were segmented into repetitions for each muscle and exercise. The data were processed as follows:As a preliminary step, the zero lines of the raw EMG data without muscle activity were processed by calculating the mean amplitude of the baseline signal. This step helped to establish a baseline for further analysis.To focus on the relevant frequency range associated with muscle activity, a band-pass filter was applied with a high-pass cutoff frequency of 20 Hz and a low-pass cutoff frequency of 450 Hz. This step eliminated unwanted noise and frequencies outside the range of interest.Following band-pass filtering, the EMG signal was rectified.To further refine the processed EMG data and reduce high-frequency noise, a low-pass filter with a cutoff frequency of 6 Hz was applied. This smoothing step helped to create a more stable representation of muscle activation patterns for analysis.To facilitate a meaningful comparison of muscle activation patterns, the data were normalized to the highest peak observed in the exercise with the greatest moment generation for each individual.

In Python (version 3.10), the collected data were averaged and interpolated using linear interpolation to generate a uniform timeline representation with consistent data points at 100% intervals. Furthermore, we used two-way ANOVA to compare how different exercises affect muscle forces.

## 3. Results

Generally, an increase in weight led to higher moments, which were particularly evident in the back squat, where 50% of body weight was applied, and in the single weight-bearing front leg during the split squat. The highest moments (2.21 Nm/kg) were observed in the rear leg due to its longer moment arm. However, in the case of good mornings, there was little variation (<0.11 Nm/kg) in moment across different angles. The maximum knee flexion angles during the front leg and back squats were found to be similar. Notably, during good mornings, participants did not maintain completely straight legs; instead, there was a slight bend of approximately 40° (Table 1). Additionally, the initial knee angles in the split squat varied from zero due to the starting position.

### 3.1. Gluteal Muscles

During exercises such as the back squat, good mornings and the front leg of the split squat, notably higher forces were observed in the intermediate part of the m. gluteus maximus compared to its medial and lateral counterparts. The peak forces occurred at the maximum knee flexion angle. Notably, the addition of extra weight to the back squat did not significantly impact the muscle forces. The medial part of the m. gluteus medius was most active when the motion was started in an upright position. It remained active during the ending standing phase as well. In contrast, the lateral part showed a slight increase in activity during the motion. In the back leg of the split squat, the m. gluteus medius medial part generated force at the beginning and end of the motion, while other gluteal parts remained mostly inactive (Figure 2).

### 3.2. Quadriceps Muscles

The various exercises had distinct effects on the quadriceps muscles. Notably, the m. rectus femoris remained consistently active throughout all exercises, except in the front leg of the split squat. In the case of the m. rectus femoris, its force steadily increased with rising knee angles. A similar trend was observed for the m. vastus lateralis and m. vastus medialis but at comparatively lower force levels. During good mornings, force increased only slightly with the elevation of knee angles. Conversely, the front leg of the split squat exhibited minimal activation in these muscles (Figure 2). Interestingly, the addition of more weight did not significantly impact the maximum force (26 N/kg) observed during the back squat, which aligns with the results seen in the gluteal muscle group.

### 3.3. Hamstring Muscles

In comparison to the other muscle groups, the hamstring muscles exhibited lower levels of muscle force. Notably, the m. semitendinosus remained nearly inactive during all exercises (<5 N/kg).

The highest activation of the hamstring muscles was observed in the front leg of the split squat, where the m. semimembranosus and m. biceps femoris reached their peak at maximum knee flexion. The m. biceps femoris short head showed slight activity at the beginning and end of the motion. A similar pattern was observed for the m. biceps femoris short head in the other exercises, such as the back squat and good mornings. During the back squat, the m. semimembranosus and m. biceps femoris long head also exhibited increased forces (<15 N/kg) with rising knee angles (Figure 2).

### 3.4. Statistics

The results showed that exercise type has an impact on muscle groups, with a level of significance close to 0.05 (F-statistic = 2.53, *p* = 0.0503), as seen in Table 2. Additionally, we found a significant difference in how exercises affect individual muscles (F-statistic = 4.63, *p* < 0.0001). In simpler terms, both the choice of exercise and muscle group matter, but muscle selection appears to have a stronger influence.

## 4. Discussion

This pilot study was conducted as part of a broader research project aimed at enhancing the safety and effectiveness of strength training through the integration of mobile technology. In this particular phase of the study, we concentrated on three distinct strength exercises: the back squat, split squat and good mornings. We used advanced computer modeling software to specifically investigate factors such as muscle forces and joint angles during these exercises. Our objective was to gain preliminary insights into how muscle activity and joint angles interact during these exercises, thereby enhancing our understanding of the internal biomechanical conditions specific to women.

### 4.1. Gluteal Muscles

The gluteal muscle group has emerged as a critical player in countering dynamic knee valgus collapse, primarily through the hip abductor function of the m. gluteus medius, as highlighted by Maniar et al. [5]. Because women tend to generate lower muscle forces in their hip abductor muscles, they face a heightened risk of excessive knee abduction, which serves as a significant contributor to the risk of ACL injuries [20]. In our study, we consistently observed that the m. gluteus medius played a significant role in all of the exercises examined: the back squat, split squat and good mornings. This emphasizes the crucial contribution of the m. gluteus medius muscle group to these exercises. The extent of the role of the m. gluteus maximus in influencing ACL injury risk remains incompletely understood. Notably, a study conducted by Alkjaer et al. [26] provided compelling evidence that the m. gluteus maximus generates posterior shear forces at the tibia, potentially contributing to a reduced likelihood of knee injuries. However, it is essential to acknowledge some differences between our findings and those of Schellenberg et al. [21] regarding the m. gluteus maximus. Specifically, Schellenberg et al. reported much lower m. gluteus maximus force during good mornings compared to our results. In Schellenberg et al., participants executed the good morning exercise with straight legs, meaning there was less than 10° of knee flexion. In contrast, our research involved a different approach, with a knee angle of approximately 39.8° during the good mornings. This difference in technique led to the activation of different muscle groups. Our data clearly show that a greater knee flexion angle results in significantly higher force generation in the intermediate part of the m. gluteus maximus. Indeed, we observed a similar trend in the front leg of the split squat, where greater knee flexion angles were associated with higher forces generated in the intermediate part of the m. gluteus maximus.

### 4.2. Quadriceps Muscles

Females often exhibit lower muscle strength and distinct activation patterns in their lower leg muscles. When facing anterior tibial translation forces during both pre-planned and unplanned athletic tasks, they often prioritize recruitment of the quadriceps muscles as their primary response [20]. In our research, we observed that quadriceps muscle activation was prominent across all exercises, with a consistently high level of force production. This suggests that exercises like the back squat primarily target the quadriceps muscles, especially the m. rectus femoris. In our study, the rectus femoris has shown a large muscle force and a large EMG activity. This is in agreement with the results from different squat exercises [27,28,29]. The EMG activity of the rectus femoris is also dependent on the type of exercises; here, it seems to play a role if the exercise is multi- or single-joint [30].

Conversely, in the split squat, we noticed that the back leg predominantly recruited the quadriceps muscles, with minimal involvement of other muscle groups, all at a high force exceeding 15 N/kg. These findings contrast with those of Schellenberg et al. [21], who reported elevated forces mainly in the vasti muscles, although the m. rectus femoris remained consistently active throughout the movement. The variation in outcomes might be attributed to differences in training experience, which in turn lead to distinct muscle control patterns. In our data set, subjects seemed to rely more on the back leg during both the descent and ascent phases, while the front leg played a stabilizing and descent-controlling role.

In the case of good mornings, we observed constant activation of the m. rectus femoris throughout the movement. This heightened quadriceps activation can be attributed to the higher knee angle of approximately 39° maintained during the exercise.

Given that the quadriceps significantly contribute to loading the ACL and play a role in anterior tibial translation, an elevated recruitment of the quadriceps during athletic tasks can heighten the risk of ACL injury [5]. Therefore, based on our findings, exercises that predominantly target the quadriceps group may not be ideal, particularly the back squat. Such exercises could potentially increase the risk of ACL injury due to their emphasis on quadriceps activation. It is advisable to incorporate a more balanced approach to strength training and conditioning or to change the exercise techniques.

### 4.3. Hamstring Muscles

Studies [31,32,33] indicate the changeable nature of the H/Q (hamstring/quadriceps) ratio, which favors the hamstring muscles due to their potential to unload the ACL and generate posterior shear force at the tibia. This effect is dependent on the knee angle, with the hamstring muscles having the greatest impact on ACL unloading at knee angles of 20° to 30° flexion. Among the hamstring group, the biceps femoris group has the strongest effect on ACL unloading. In comparison to the m. biceps femoris, the orientation of the m. semimembranosus limits its potential for unloading [5].

Based on our results, it appears that the hamstrings exhibit lower activation levels during these exercises. In contrast to the knee angle range proposed by Maniar et al. [5], our findings indicate that the hamstrings generate force at the beginning of the movement (0° to 40°) and at the maximum knee flexion angle (100° to 120°). Specifically, the m. biceps femoris short head exhibits activation in the range of 20° to 30°. The force production of this muscle was consistently observed in back squats, in good mornings and in the front leg of split squats, all at approximately 10 N/kg.

The m. semimembranosus muscle begins to produce force at the end of the movement cycle at the maximum knee angle during back squats and in the front leg portion of split squats. However, there is minimal activation during good mornings and split squats with the back leg. In comparison to Schellenberg et al. [21], our results align during the split squat. The m. semimembranosus and m. biceps femoris long head muscles exhibited peak force production at a knee angle of 80°, which is lower than our observed range.

In the context of good mornings, our results show activation only in the m. biceps femoris short head within the specified angle range. Schellenberg et al. [21] conducted the exercise below the most active angle range for ACL unloading. In an extended leg position, the hamstrings have a minimal effect on ACL unloading due to their line of action in this position, as noted by Maniar et al. [5]. Therefore, while the execution of good mornings targets the hamstrings more effectively at higher force levels, it trains the muscles at an angle with little contribution for ACL unloading.

To effectively target the hamstrings, consideration must be given to the desired range. Our study suggests that good mornings could be performed with a smaller knee angle and split squats with a limited angle range to more effectively achieve the desired shift in the H/Q ratio.

### 4.4. Comparison to EMG Activation

Training recommendations were previously established based on EMG muscle activity patterns [10,11,12,14,34]. Our measured EMG patterns exhibited a similar activation pattern during the back squat, with the quadriceps group showing the highest level of activation. Specifically, the vasti muscles demonstrated the most significant EMG activity, consistent with the findings of Saito et al. [13,14]. In the case of the split squat, we observed a slightly greater activation of the vastus lateralis compared to the vastus medialis, which contrasts with the results reported by Irish et al. [35]. When comparing the force patterns of the musculoskeletal model to the measured EMG patterns, we identified similarities in the targeted muscle groups. The quadriceps were the primary focus in the back squat, while the vasti and hamstrings were the key muscle groups involved in the good morning exercise. However, discrepancies emerged in the split squat. EMG indicated higher activation of the quadriceps muscle group in both the front and back legs, whereas the modeled forces tended to emphasize the hamstring muscles in the front leg (Figure 3).

EMG is a valuable tool for understanding muscle activation patterns, but it comes with certain limitations. These limitations include cross-talk between muscles, the importance of precise electrode placement and the fact that it does not directly measure muscle force [17,36]. Given the known limitations of EMG in accurately assessing muscle force magnitude, our results stress that the evaluation of dynamic strength training exercises only through surface EMG measurements may have its restrictions. Lorenzetti et al. [16] have also confirmed these findings. Their research findings emphasize that recommendations for muscle force and magnitude cannot only rely on EMG data and cannot be directly inferred from the corresponding force and joint moments.

### 4.5. Limitations

Measurements of kinematic data were taken during different sessions with only eight subjects. Therefore, the measurement setup and each individual’s exercise execution had a significant influence on subsequent modeling processes. The small sample size of eight female participants in our study limits its generalizability and applicability. This constraint restricts the ability to extrapolate the findings to a broader population, making it crucial to interpret the results with caution. However, despite these limitations, the insights gained from this initial cohort offer valuable preliminary evidence on the effects and applicability of the proposed. It underscores the need for subsequent research to incorporate larger and more diverse samples, thereby enhancing the robustness and external validity of the findings. During the data acquisition phase, the accuracy of the ground reaction forces measured was compromised by the necessity of manual adjustments (rotation of 180 degrees) and hence adjustment within the Nexus software (v2.14), a step that potentially introduces errors, although such inaccuracies were unforeseen. Precise experimental data are crucial for accurately estimating muscle forces in the model. Mistakes in joint movements and torque calculations, for instance, would greatly affect the estimation [37].

Musculoskeletal modeling for assessing internal loading relies on various assumptions and simplifications, such as subject-specific anatomy and physiology, the determination of muscle origin and its line of action, as well as its cross-sectional area and the path of muscles relative to the skeleton. These factors collectively determine the moment arms for each muscle, which in turn define the linear force–length relationships. However, it is important to note that these relationships may not fully capture the complexity of muscle physiology. Predefined maximum isometric force values and estimated physiological cross-sectional areas of muscles have been shown to impact the magnitude of muscle force estimates, particularly with the inverse dynamics–static optimization method, as used in our study [37]. Lower cross-sectional area values are associated with higher muscle force estimates. Therefore, these estimated parameters may not completely account for individual variations in muscle properties.

There is currently no validated model for strength training involving higher loads and deep knee flexion. In most cases, to validate muscle force predictions, researchers typically compare muscle loading or activation patterns with EMG data as a measure of validity. However, it is important to note that, while analysis of the timing and intensity of muscle activation during a movement is helpful, such comparisons cannot directly confirm the accuracy of the calculated muscle force magnitude. Therefore, the generic models commonly used may not be entirely applicable to motions involving various knee angles. The Gait2392 model used here is particularly suitable for analyzing knee flexion/extension within the range of −99° to 0°. A study conducted by Schellenberg et al. [38] revealed significant errors during squats, especially at the deepest knee flexion angles, where internal loading conditions led to a substantial 60% peak error. This error resulted in an overestimation of loading at the maximum knee angle and an underestimation at the minimum flexion.

These limitations serve as a reminder to approach the precise data with caution. Nonetheless, the data provide valuable insights into how the internal loading of particular muscle groups can impact knee biomechanics. Future studies should prioritize the validation of the model across a complete range of motion for the specific exercise and incorporate subject-specific model parameters:Development of a scaled model based on magnetic resonance imaging by determining muscle origins, calculating cross-sectional areas from muscle volume and bone geometries derived from the images and establishing their fCoRs [39].Establish a precise setup and validate the data with a Vicon model to ensure accurate measurements of joint angles and moments in the desired joint, particularly because static optimization relies on the foundation provided by inverse dynamics.Increase the sample size by including more participants. This will enhance the validity of the study and improve the overall reliability of the data.

## 5. Conclusions

Our primary aim was to explore the impact of various strength exercises on muscle activity and joint angles in women, thereby helping to make strength training safer and more effective, with a particular focus on the ACL. We used models and EMG measurements to understand how muscles, joint angles and movements impact the ACL during daily activities and sports. As expected, our study found that the alignment between modeled muscle forces and EMG measurements varies across different muscle groups and exercises. Additionally, muscle forces depend on the chosen exercise and the specific range of joint angles used for training. Our results consistently showed greater involvement of the quadriceps muscles in most exercises. Conversely, the hamstrings produced lower force levels during these exercises. According to the existing literature on ACL injury prevention (particularly at joint angles of 20°–40°, ), this range appears to be ideal for training the hamstrings. This alignment was observed during good mornings and the back leg portion of the split squat, where hamstring forces were highest within this angle range. It is important to note that our study has limitations related to data collection setup and the musculoskeletal model used. Moreover, surface EMG data have their own restrictions. Nonetheless, while the absolute values in our study should be interpreted with caution, they do provide valuable insights into muscle behavior during different exercises. This can be useful for making exercise recommendations. Furthermore, we learned that the creation of custom models for specific exercises and individual characteristics can provide better force data. This approach enhances our understanding of the effects of various exercises on muscular interplay and lays the groundwork for future research aimed at improving the safety and efficacy of strength training.

## Figures and Tables

**Figure 1 jfmk-09-00068-f001:**
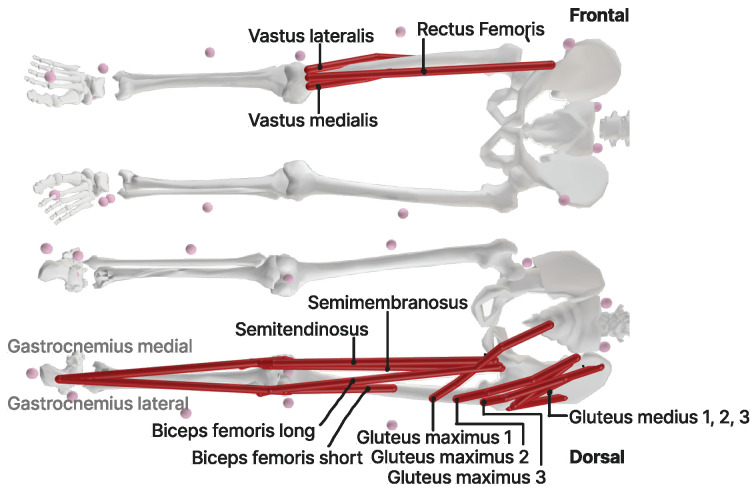
OpenSim muscles evaluated. (Frontal) Quadriceps muscles: rectus femoris, vastus lateralis and medialis. (Dorsal) Gluteal muscles: M. gluteus maximus and m. gluteus medius, where m1 = medial part, m2 = intermediate part and m3 = lateral part. M. gastrocnemius lateralis and medialis and hamstring muscles: m. semitendinosus, m. semimembranosus, m. biceps femoris long and short head.

**Figure 2 jfmk-09-00068-f002:**
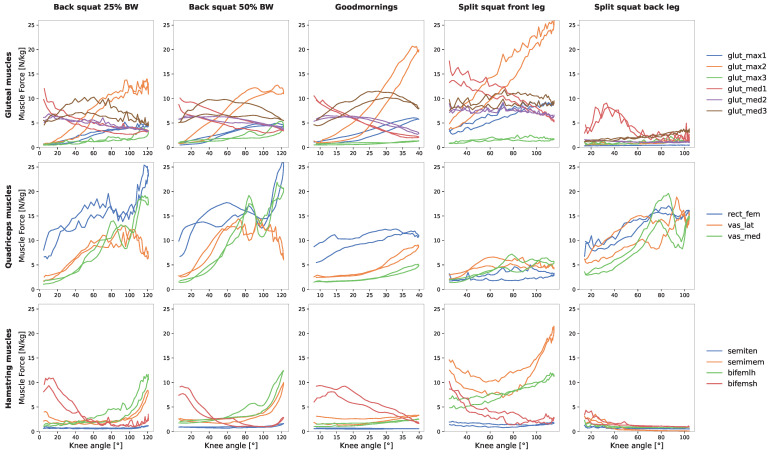
Comparative analysis of normalized mean muscle forces (relative to participants’ body weight) in relation to knee angles. The m. gluteus maximus and m. gluteus medius are characterized as follows: m1 = medial part, m2 = intermediate part and m3 = lateral part.

**Figure 3 jfmk-09-00068-f003:**
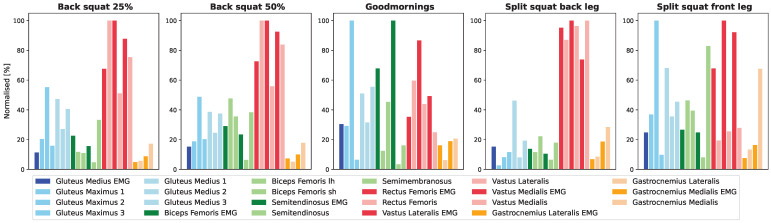
Illustration of the relative impacts of different exercises on muscle forces and EMG measurements, both scaled to 100%. Bright colors represent the EMG Max values, while pastel colors represent the maximum musculoskeletal modeled forces. The color key is as follows: blue for gluteal muscles, green for hamstring muscles, red for quadriceps muscles and yellow for gastrocnemius muscles.

**Table 1 jfmk-09-00068-t001:** Maximal normalized EMG activation, knee angles and moments alongside their respective standard deviations across the exercise variations.

	Back Squat 25%		Back Squat 50%		Good Mornings		Split Squat Front Leg		Split Squat Rear Leg	
	Mean	sd	Mean	sd	Mean	sd	Mean	sd	Mean	sd
Angles [°]	120.7	13.5	122.3	14.6	39.8	11.03	115.19	6.7	104	11.5
Moments [Nm/kg]	1.05	0.2	1.29	0.3	0.11	0.02	1.52	0.6	2.21	0.6

**Table 2 jfmk-09-00068-t002:** Impacts of different exercises on muscle groups. Rows represent the muscles studied while columns represent the exercises performed. The table helps to identify which exercises significantly affect specific muscles and explains the sources of variation in the data.

ANOVA						
Source of Variation	Sum of Squares	df	MS	F-Value	*p*-Value	Critical F-Value
Rows	1621.87	14	115.85	4.63	$1.75\times 10^{-5}$	1.87
Columns	253.69	4	63.42	2.53	0.05	2.53

## Data Availability

Data contained within the article.

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
