# Peer review of "Female Lower Body Muscle Forces: A Musculoskeletal Modeling Comparison of Back Squats, Split Squats and Good Mornings"

_jfmk, 2024, doi:10.3390/jfmk9020068_

Round 1
Reviewer 1 Report
Comments and Suggestions for Authors
The purpose of the current study was to analyze the muscle forces produced during various strength training exercises (back squats, good mornings, split squats) of the lower body in females. This was done to provide information that may be useful for ACL injury prevention in females. A total of 8 subjects were studied with a barbell load of 25% to 50% of bodyweight (back squat only) in the strength training exercises. Biomechanical analyses (kinematics, kinetics) were performed during performance of the exercises The main findings were that the back squat required the highest quadriceps forces and back led during the split squat, whereas the gluteal muscle forces were highest in the good morning and the front leg of the split squat. For the hamstrings, the highest muscle forces were in the front leg of the split squat.
Overall, the design of the study was pretty good and the topic studied is important. The study was also well-written. I think the study would definitely be of interest to the readers of JFMK. I don’t think the study has any fatal flaws. However, I do have a number of questions and possible corrections that the authors need to address. More information on several aspects has to be added.
1The author’s acknowledged the limitation of only having 8 subjects, which is appreciated. Was there any sort of a priori power analyze done? More concerning is in the limitations paragraph the statement that “the individual’s exercise execution had a significant influence on subsequent modelling processes. In our setup, the accuracy of the ground reaction forces was not completely precise” I don’t understand why the GRF would not be precise. The statement also implies that the subject’s technique wasn’t good?? Please explain.
2Almost no information is given about the details of the exercise execution. Were these half squats or deep full squats, was the depth controlled for individual subjects, was it different across subjects, what were the details of the other exercises. Obviously the depth of the squats would affect all of those calculations right?
Good Morning should be two words throughout not goodmorning.
Why was a percentage of bodyweight chosen as the load instead of a 1RM or at least an estimate of 1RM based on a repetition test?
The EMG processing. What does it mean the raw EMG were processed by calculating the mean? The mean of the interference EMG??? I assume not but then afterward it says it was rectified. I am not sure what mean would be done before rectification.
It is surprising that rectus femoris was highly activated in the back squat several studies have shown since it is a two joint muscle it is not overly active in squats but is in leg extensions. This issue should at least be discussed.
Line 187 “moderate level of significance” please change this wording
The table right before the discussion is 7 decimal points really needed? I would think the journal would say two is the most needed.
Lines 235 and 236 imply that people did the exercises with different technique, please explain.
Line 277 I don’t understand what incorrect line of action means?
Bibliography has errors in the capitalization of the first letters of the words of the article titles, some references have them capitalized others don’t.
Comments on the Quality of English Language
minor proofreading
Reviewer 2 Report
Comments and Suggestions for Authors
The aim of this study was to analyze muscle forces in the lower leg during strength exercises such as back squats, goodmornings and split squats. The authors focused on women, who are more vulnerable to anterior cruciate ligament injury, in order to better understand muscle engagement and its role in injury prevention. The variation in muscle forces in these exercises was assessed and their alignment with measured EMG was investigated. Their implications for ACL injury risk factors were also studied and discussed.
The subject is interesting and fits in perfectly with the journal's themes. The article is well structured, the statistical method appropriate and the results relevant. The discussion is interesting and a convincing limitation section is proposed. The originality of the method should be highlighted and detailed in relation to solutions proposed in the international literature. I think it is important to consider the following questions and comments in order to improve the article:
- In the "method" section: the sample of women is small (8 subjects), what are the implications for the generalizability and applicability of the proposed approach? This point is very important and needs to be explained and detailed to demonstrate the relevance of the work proposed to readers. These points deserve to be commented on and limitations added to the discussion, particularly with regard to the generalization of the process.
- In the "data collection and items" section. The choice of muscles should be discussed and commented on in relation to the literature, the activity studied and the objective of the article. Justification of muscle selection is essential.
- Section 2.2: The authors use a VICON optoelectronic system. The positioning of the cameras should be explained and justified. In addition, the biomechanical parameters quantified with the optoelectronic system should be described, as well as how they are obtained (computed) from the test patterns.
- A brief description of EMG positioning and the precautions taken to ensure good quality of the signals measured could be added.
- In the Kinematics and kinetics section: How were the benchmarks defined? What convention was used? Why not use the ISB conventions conventionally used in this type of work?
- The authors write on line 139-140 "in Python the collected data were averaged and interpolated to generate...", which interpolation method is used?
Round 2
Reviewer 1 Report
Comments and Suggestions for Authors
The authors have done a good job of answered all of my previous comments. It seems that the limitations will be addressed in future work. They implemented the changes I requested in most all cases.
Author Response
Dear Reviewer,
We sincerely appreciate your insightful feedback on our manuscript.
Thank you and kind regards, Basil Achermann
Reviewer 2 Report
Comments and Suggestions for Authors
The majority of the questions asked were commented on, but the manuscript was not modified to incorporate the recommendations and improvements made during the first review. These modifications and improvements are necessary to consider the article for publication.
Author Response
Dear Reviewer,
We sincerely appreciate your insightful feedback on our manuscript. Following your suggestions, we have made the necessary amendments, which are highlighted in the attached PDF file for your review.
We are grateful for the time and effort you have dedicated to enhancing our work.
Kind regards, Basil Achermann

Round 3
Reviewer 2 Report
Comments and Suggestions for Authors
Thanks to the authors for taking the comments and recommendations into account.